# Advances in Micro-/Mesopore Regulation Methods for Plant-Derived Carbon Materials

**DOI:** 10.3390/polym14204261

**Published:** 2022-10-11

**Authors:** Jing Liu, Ke Zhang, Huiyan Wang, Lin Lin, Jian Zhang, Peng Li, Qiang Zhang, Junyou Shi, Hang Cui

**Affiliations:** 1Key Laboratory of Wooden Materials Science and Engineering of Jilin Province, Beihua University, Jilin 132013, China; 2Beijing Spacecraft Manufacturing Co., Ltd., Beijing 100094, China; 3School of Mechanical Engineering, Yanshan University, Qinhuangdao 066004, China; 4National Demonstration Center for Experimental Physics Education, College of Physics, Jilin University, Changchun 130012, China

**Keywords:** plant, porous carbon, pore size, microstructure

## Abstract

In recent years, renewable and clean energy has become increasingly important due to energy shortage and environmental pollution. Selecting plants as the carbon precursors to replace costly non-renewable energy sources causing severe pollution is a good choice. In addition, owing to their diverse microstructure and the rich chemical composition, plant-based carbon materials are widely used in many fields. However, some of the plant-based carbon materials have the disadvantage of possessing a large percentage of macroporosity, limiting their functionality. In this paper, we first introduce two characteristics of plant-derived carbon materials: diverse microstructure and rich chemical composition. Then, we propose improvement measures to cope with a high proportion of macropores of plant-derived carbon materials. Emphatically, size regulation methods are summarized for micropores (KOH activation, foam activation, physical activation, freezing treatment, and fungal treatment) and mesopores (H_3_PO_4_ activation, enzymolysis, molten salt activation, and template method). Their advantages and disadvantages are also compared and analyzed. Finally, the paper makes suggestions on the pore structure improvement of plant-derived carbon materials.

## 1. Introduction

Carbon materials can exist in a 0D to 3D structure stably in nature due to their outstanding chemical stability. The multiple allotropes of carbon make it diverse in structure and versatile [1]. Novel carbon materials mainly include graphene [2], carbon nanotubes [3], porous carbon [4] and fullerene [5], derived from resins [6], gels [7], biomass materials [8], petroleum pitch [9], etc. In past decades, with the development of society, the demand of people for energy is increasing. Therefore, the exploitation of renewable and clean energy is particularly significant [10,11]. Biomass-based carbon with good electrical conductivity and high porosity can be obtained by high-temperature treatment (hydrothermal carbonization, pyrolysis and gasification) under oxygen-limited conditions [12]. Compared with other materials, biomass-based carbon materials have abundant precursors, such as animal waste and its derivatives, plant waste and its derivatives and microorganisms (bacteria) [13,14]. Plant materials are favored by researchers because they are more readily available in nature and have lower costs than animals and microorganisms. Carbon materials have been successfully prepared from natural plant materials (such as onion skin [15], rice husk [16], fruit peel, sawdust [17], lotus stem [18], nut shell [19] and seed [20]) and plant extracts (such as lignin [21], cellulose [22], sodium alginate [23] and glucose [24]). Some plant materials can maintain a more complete structure after carbonization and possess strong mechanical properties, which can be used directly as functional materials to simplify the preparation process. Plant-derived carbon materials are characterized by diverse microstructures and rich chemical compositions, providing the basis for their wide applications (catalysis [25], electromagnetic wave absorption [26], electrochemical energy storage [27], adsorption [28], etc.). In addition, the pore size distribution of plant-derived carbon materials is also a major factor influencing their performance. According to the definition of pores by the International Union of Pure and Applied Chemistry (IUPAC), pores are classified into micropores (<2 nm), mesopores (2–50 nm) and macropores (>50 nm) according to their pore size, with micropores providing more adsorption sites, mesopores facilitating faster transport and macropores shortening diffusion distances [29,30]. However, many plant materials suffer from a high proportion of macropores, which limits their applications. Therefore, in this paper, the advantages of plant-derived carbon materials are briefly introduced in terms of chemical composition and microstructure. Then, previous studies are reviewed to address the above-mentioned high proportion of macropores, and methods are summarized for micro-/mesopore regulation. Finally, this paper makes suggestions on the application and pore structure improvement of plant-derived carbon materials.

## 2. Advantages of Plant-Derived Carbon Materials

### 2.1. Various Chemical Compositions

Containing rich heteroatoms is one of the natural advantages of plants as a class of carbon precursors. In other words, besides C, there are many other elements, such as O, N, S, P and Si, which can boost the development of more advanced materials and functional materials. The existing forms of O in plant-derived carbon materials are mainly -OH, -COOH, C=O and C-O-C, and these functional groups can be retained in the carbon network even at high temperatures, which can enhance the wettability of the surface of carbon material and provide the pseudocapacitance for supercapacitors and lithium batteries to improve the electric capacity [31]. N atoms exist in carbon materials mainly in the form of pyridine nitrogen, pyrrole nitrogen and graphite nitrogen. In addition to enhancing the hydrophilicity of the carbon materials, they are also able to increase electron mobility [32] and form a large number of electrocatalytic active centers in the bulk structure of electrocatalysts [33]. With larger radii, smaller electronegativity and multiple covalent bonds, S atoms play unique roles in different applications. Compared to N-doped porous carbon, the S-doped counterpart has more outstanding electrical conductivity, and N-containing functional groups can help dissociate oxygen molecules in the field of electrocatalysis, while S can facilitate proton transfer [1]. Owing to the presence of P-containing functional groups, researchers can develop plant-derived carbon materials with high carboxylation, dehydration, aromatization and oligomerization activities, which are promising catalyst candidates to raise the yields of some compounds. Unlike other elements, Si is mainly present in plant materials as SiO_2_, which can act as a templating agent to create new pores after its removal by acidic reagents and thereby regulate the pore structure of plant-derived carbon materials [34]. So far, some plants have been found rich in heteroatoms, which can be directly pyrolyzed in the absence of any chemical reagent to obtain heteroatom-rich carbon materials. Gopalakrishnan et al. [14] employed onion skin as a carbon precursor to synthesize heteroatom (N, P, S) co-doped porous carbon nanosheets via a carbonization and activation strategy. Xia et al. [35] prepared a sulfur self-doped carbon material with *Camellia japonica* containing natural sulfur as a raw material and used it as an electrode material to achieve a specific capacitance of 125.42 F/g at a current density of 2 A/g. Gong et al. [36] obtained porous carbon materials by using nitrogen-containing duckweed as a raw material with KOH activation. The results showed that the self-doped nitrogen provided abundant active sites for oxygen reduction and significantly improved the oxygen reduction kinetics. In addition, many other plants have also been served as raw materials or templates to prepare self-doped and self-template carbon materials, which are summarized in Figure 1.

### 2.2. Diverse Structures

Plants show diversity in both macroscopic and microscopic structures. From the perspective of microstructure, they can be made into carbon materials with various morphologies, such as zero-dimensional (0D, sphere), one-dimensional (1D, tube), two-dimensional (2D, sheet) and three-dimensional (3D, skeleton) [50].

#### 2.2.1. Zero-Dimensional (0D) Carbon Materials

Carbon microspheres (0D) possess a unique hollow structure, which can provide a high specific surface area and a short diffusion distance. Using pomelo peel as a raw material, Zhuang et al. [51] synthesized carbon quantum dots (CQD) by the hydrothermal method. Jagannathan et al. [52] employed corn cob as a raw material to synthesize CQD and fabricated white light-emitting sheets (Figure 2), which displayed uniform brightness and had good color reproducibility and high stability under various UV light. Yan et al. [53] obtained carbon microspheres from oatmeal and employed them as the anode material of sodium ion batteries. The electrochemical results showed that the capacity remained at 104 mA/h·g^−1^ after cycles without significant decay.

#### 2.2.2. One-Dimensional (1D) Carbon Materials

Carbon nanotubes, carbon fibers and carbon microtubes are common 1D carbon materials. They can be obtained by self-assembly, hydrothermal, electrostatic spinning and the solvothermal method [54]. Wu et al. [55] prepared ZnO/porous carbon microtubes (ZnO/PCMT) by dip coating and thermal etching on sycamore microtubes, which had a high specific surface area (1076 m^2^/g) and excellent electromagnetic wave absorption capability. Long et al. [56] synthesized carbon microtubes through the high-temperature pyrolysis of kapok fibers (Figure 3a–d). As can be seen from Figure 3b’,c’, the surface microstructure of the carbonized kapok fibers was rougher than that of the uncarbonized, and performed excellently in microwave absorption with an optimized effective absorption bandwidth of 7.12 GHz (10.64–17.76 GHz, 2.3 mm). High-performance carbon fibers are mostly generated from petroleum-based materials. However, the preparation methods are expensive and can cause environmental degradation. As a result, low-cost and eco-friendly plant materials are utilized in the production of carbon fiber materials. Guo et al. [57] prepared carbon fibers by hydrothermal carbonization using C. Sinensis branch waste as the raw material. Jiang et al. [58] used an alkaline solution to extract natural cellulose from bamboo chopsticks to create carbon fibers. Osman et al. [59] employed potato peel waste as a raw material to develop carbon materials, which were mixed with nitrogen-based material melamine and iron (III) oxalate hexahydrate, for preparing multi-walled carbon nanotubes with strong hydrophilicity (contact angle θ = 14.97°).

#### 2.2.3. Two-Dimensional (2D) Carbon Materials

Graphene, as a 2D carbon material, has high electrical conductivity and strong stability. Chemical vapor deposition (CVD) and graphite stripping are two common methods to prepare graphene. However, the high cost of CVD and the non-renewability of graphite exfoliation are not conducive to sustainable development [60,61]. Therefore, the preparation of graphene materials using plants as raw materials has been explored. Laser induction is used to prepare graphene because of its simple operation. Ye et al. [62] investigated the effect of CO_2_ laser power on the quality of the generated graphene when using pine wood as the raw material (Figure 4a), the pattern of pine etched by the CO_2_ laser was displayed in Figure 4b. And the SEM (Figure 4c,d) and TEM (Figure 4e,f) of P-LIG-70 showed that the CO_2_ laser-treated wood produced layered porous graphene structures. Mahmood et al. [63] directly synthesized lignin-derived graphene at different laser powers. This graphene material had the smallest impedance at the laser power of 80%, and better electrochemical performance (90% capacitance retention after cycles) was obtained by assembling it into flexible supercapacitors.

#### 2.2.4. Three-Dimensional (3D) Carbon Materials

Hierarchical porous carbon materials have a 3D carbon skeleton structure and usually contain two or three kinds of pores of different sizes, i.e., micropore–mesopore, micropore–macropore, mesopore–macropore and micropore–mesopore–macropore. Because of more active sites, a short diffusion distance and a higher transport rate, hierarchically porous carbon materials are favored in the field of electrochemical energy storage. Chen et al. [64] obtained 3D hierarchically porous carbon (3DHC) through high-temperature pyrolysis and compounded it with NiCo_2_S_4_ nanowires by the hydrothermal method. When used as an electrode material, the NiCo_2_S_4_/3DHC possessed specific capacitance as high as 765.8 F/g at a current density of 1 A/g. Subramanian et al. [20] used *Datura stramonium* seed pods as a raw material to develop a novel carbon material with the help of high-temperature carbonization, which had a 3D interconnected network structure and a specific surface area up to 1390 m^2^/g. Wei et al. [65] obtained 3D wood-derived carbon materials from waste wood by direct pyrolysis and assembled them with polyaniline-modified 3D wood-derived carbon electrode materials to form a desalination device (Figure 5). In summary, carbon materials of different dimensions have different characteristics; therefore, the combination of different dimensions of carbon materials to diversify their functions is a major current research trend.

## 3. Pore Size Regulation Methods

### 3.1. Micropore Regulation Methods

#### 3.1.1. Alkali Activation

Generally, alkali activation improves the pore distribution. KOH and NaOH are common alkaline activators. Differently, the K atom dissociated from KOH can embed into all materials, while the Na atom sourced from NaOH is only for the very disordered materials [66]. The activation mechanism of KOH is shown in Equations (1)–(6): KOH first reacts with C to produce K, H_2_ and CO_2_; when the temperature up to about 400 °C, K_2_CO_3_ is generated. At about 600 °C, KOH is completely consumed; when the temperature reaches 700 °C, K_2_CO_3_ is decomposed to produce K_2_O and CO_2_, and K_2_CO_3_ and K_2_O are reduced by C to produce metal K. When the temperature reaches 800 °C, K_2_CO_3_ is completely decomposed. In the process of the above reaction, the carbon material is etched by the resulting gas to produce more pore structures. Moreover, the metal K is embedded in the carbon lattice, which expands to form pores [67,68].
KOH + C → 2K + H_2_ + CO_2_(1)
2KOH + CO_2_ → K_2_CO_3_ + H_2_O(2)
K_2_CO_3_ → CO_2_ + K_2_O(3)
CO_2_ + C → 2CO(4)
K_2_CO_3_ + 2C → 2K + 3CO(5)
C + K_2_O → 2K + CO(6)

Researchers investigated the impact of KOH on the pore structure of various plant materials and found that KOH can effectively prepare materials dominated by micropores. Some studies of activating plant-based carbon materials with KOH are listed in Table 1.

Okonkwo et al. [74] employed castor shells to prepare a novel carbon material with a microporosity of 85% through KOH activation at 800 °C. The research groups of Shen and Souza explored the effects of temperature and the char/alkali mass ratio on microporosity. Shen et al. [70] obtained porous carbon by mixing orange peel with KOH at a mass ratio of 1:1. The microporosity of the material increased to 70% when the temperature rose from 600 to 800 °C and decreased to 45% as the temperature reached 900 °C (Figure 6a–c). Moreover, the specific surface area and pore volume reached the maximum at 800 °C, which indicated that 800 °C was the optimal activation temperature. Souza et al. [78] selected biomass waste as a raw material for the sustainable production of nanoporous carbon and explored the influence of KOH content on the activation effect. As the alkali content increased, the specific surface area and pore volume became higher, while the microporosity decreased (Figure 6d–f), which was ascribed to excessive alkali making micropores merge to form mesopores. The above studies demonstrate that KOH activation is an effective means of regulating the microporosity of plant-derived carbon materials and that high specific surface areas can be obtained.

#### 3.1.2. Foam Activation

NaHCO_3_ and KHCO_3_ are commonly used foam activators that reduce environmental pollution. KHCO_3_ is able to serve as a substitute for KOH due to being less corrosive [79,80]. In the case of NaHCO_3_, the activation mechanism is shown in Equations (7)–(11) [81]:2NaHCO_3_ → Na_2_CO_3_ + H_2_O + CO_2_
(7)
Na_2_CO_3_ → Na_2_O + CO_2_(8)
Na_2_CO_3_ +2C → 2Na + 3CO(9)
Na_2_O + C → 2Na + CO(10)
CO_2_ + C → 2CO(11)

Liu et al. [82] synthesized sponge-like carbon materials by a one-step method using Spirulina platensis as raw material and NaHCO_3_ as the activator; the impact of temperature on the activation effect has been investigated. The results showed that the carbon materials prepared at different temperatures were dominated by micropores, and at the pyrolysis temperature of 800 °C, the specific surface area and pore volume was increased to 1511 m^2^/g and 0.93 cm^3^/g, respectively. Liang et al. [83] also explored the temperature influence on the activation effect; the microporosity of carbon material grew from 85% to 87%, and the specific surface area increased by 2.2 times when the temperature was raised from 550 to 750 °C. Yang et al. [84] prepared a porous carbon material by mixing corn straw with NaHCO_3_ and heating treatment. It can be seen from Figure 7a–c, the as-prepared carbon material possessed a large specific surface area (1891.9 m^2^/g) and high microporosity (82.3%) at 800 °C, which is 169 and 12.8 times that of untreated corn straw, respectively. Xiao et al. [85] developed hollow-tubular porous carbon by using biomass Cycas fluff (CF) through carbonization and NaHCO_3_ activation at different temperatures. As shown in the Figure 7d-f, under the pyrolysis temperature of 700 °C, porous carbon had a specific surface area of 397.92 m^2^/g and the microporosity of 73.91% at the mass ratio of 1:2 (CF:NaHCO_3_). Although the foam activation mechanism of NaHCO_3_ and KHCO_3_ is similar to that of KOH, the effect is more mild, which is beneficial to maintain microporous structures at high activation temperatures.

#### 3.1.3. Physical Activation

CO_2_ and H_2_O(g) are commonly utilized for single-step or multi-step physical activation. Single-step activation means that the material is carbonized in a nitrogen atmosphere and then directly passed into the H_2_O(g) or CO_2_ atmosphere for activation. Multi-step activation means that the secondarily carbonization in the H_2_O(g) or CO_2_ environment. Compared with multi-step activation, single-step activation has the advantages of low energy consumption and time saving [86]. The essence of this strategy is the redox reaction between the C element in the carbon material and the other elements in the oxidizing gases [87]. Table 2 shows the application of the physical activation method in plant-based carbon materials.

Chen et al. [95] reported a CO_2_ and H_2_O(g) self-activation method to prepare wood-derived carbon material under the pyrolysis temperature of 1000 °C and the gas flow rate of 10 mL/min. As shown in Figure 8a, the specific surface area and microporosity were up to 1145 m^2^/g and 50.3%, respectively. Ding et al. [87] investigated the effect of different preparation processes on the properties of the materials and found that the material subjected to multi-step activation displayed higher specific surface area (1690 m^2^/g) and pore volume (0.93 cm^3^/g), while one-step activation yielded a relatively high microporosity (72.0%), which is about 1700, 162, and 72 times larger than that of the unactivated materials, respectively. Phothong [89] and Khuong [90] investigated the effect of CO_2_ activation temperature and activation time on the pore structure of bamboo-derived carbon materials, respectively. The results showed (Figure 8b–d) that the carbon material had the maximum microporosity when the activation temperature was 850 °C. When the temperature was increased to 950 °C, the microporosity decreased to 87.8%, but the specific surface area increased to 810 m^2^/g, which was attributed to the collapse of the partial pore structures and the merging of the micropores to mesopores. Furthermore, as shown in Figure 8e–g, at the pyrolysis temperature of 800 °C, a significant effect of activation time on specific surface area and pore volume can be observed; when the activation time was 4 h, the specific surface area (1496 m^2^/g) and pore volume (0.64 cm^3^/g) reached the maximum, while the change in microporosity is smaller. Compared with KOH activation and foam activation, physical activation can take the advantage of the water contained in the material as the activator, providing a simple and eco-friendly way to regulate microporosity. However, this activation method still suffers from a relatively long activation time and large energy consumption. The effect of the gas flow rate on the experimental results needs to be considered also.

#### 3.1.4. Freezing Treatment

For plant materials, freezing treatment is mainly divided into the freeze–thaw method and the freeze-drying method. Freeze–thaw treatment places a material containing liquid in an environment at a temperature below the freezing point of the liquid and leverage the expansion stress generated by the formation of ice from the liquid to destroy the microstructure of the material to form a new pore structure. Freeze drying refers to a frozen material being placed in a vacuum freeze-dryer, and the ice in the material is directly vaporized to form pores. Compared with freeze–thaw treatment, freeze drying is in favor of better maintaining the morphology of materials. Freeze–thaw treatment can adjust the porosity of a material by changing the freezing time, the number of freeze–thaw cycles and the freezing temperature [96].

Hou et al. [97] carbonized freeze-dried lotus as a carbon precursor to prepare multi-porous biochar with outstanding microporosity (76.0%). In addition, Wang et al. [98] further explored the influence of the number of freezing times on the freezing treatment effect (Figure 9). It is found that the freezing pretreatment for three times possessed larger specific surface area, microporosity and pore volume, namely that the increase in the number of freeze–thaw cycles was beneficial to form more pores. Compared with KOH activation, foam activation and CO_2_ and H_2_O(g) activation, freezing pretreatment can effectively improve the microporosity of materials, but the specific surface areas of the resulting samples are relatively small, which is normally combined with other methods to further improve the pore structure of materials so that their performance can be improved in various fields. However, this method is less utilized in the pore size regulation of plant-derived carbon materials. Furthermore, the impacts of the freeze–thaw temperature, frequency and time on pore size regulation remain to be further investigated.

#### 3.1.5. Fungal Treatment

Plant cells are mainly composed of cellulose, hemicellulose, and lignin. Fungi are capable of regulating the pore sizes of plant materials mainly through degrading their cell structures with exoenzymes to produce new pores. Moreover, hyphae formed in the growth process of fungi can penetrate cell walls to fabricate new pores and cross-link with each other to construct a network structure, thus reducing macroporosity and increasing microporosity and mesoporosity. The process of regulating the pore structure of biomass materials by fungi is shown in Figure 10a [99].

Zhang et al. [100] took lotus leaves as the carbon precursor to prepare activated carbon and investigated the effects of different components (lignin, cellulose and hemicellulose) on the pore structure. The results indicated that the cellulose of lotus leaves was degraded by *Trichoderma viride* (*T. viride*), the specific surface area, pore volume and microporosity were 2290 m^2^/g, 1.13 cm^3^/g and 56.6%, respectively. It is suggested that the cellulose had a significant impact on microporosity. In addition, the AFM results (Figure 10b–e) show that the pretreated samples possess more regular needle-like protrusions and favor the formation of micropores. Wang et al. [101] reported a carbon material made from fungus-treated basswood by pyrolysis at 1000 °C. Compared with the pristine basswood (716.9 m^2^/g), fungus-degraded basswood had a larger specific surface area (1041.6 m^2^/g). Some researchers further investigated the contribution of fungal treatment to the microporosity of plant-derived carbon materials. Cheng et al. [102] combined the rice husks modified by *T. viride* with KOH to improve pore structure. The rice husks witnessed increased specific surface area (3714 m^2^/g), pore volume (2.07 cm^3^/g) and microporosity (50.7%) after 20 days of *T. viride* pretreatment. Moreover, the crystallinity of rice husks was enhanced after the microbial pretreatment, which is advantageous to the carbon yield. Evidently, the fungal treatment can effectively improve the microporosity of materials, with its regulation effect influenced by fungal treatment time. For wood/bamboo materials, a process of fungal treatment that is too long will lead to a loose structure that is easy to break; thus, it is not conducive to the overall utilization of the materials. However, at present, the research on utilizing fungal to regulate pore size is not common; more influence factors need to be further explored.

### 3.2. Mesopore Regulation Methods

#### 3.2.1. H_3_PO_4_ Activation

Phosphoric acid (H_3_PO_4_) has a moderate acid strength, possessing the advantages of less pollution and low energy consumption and is milder than many other chemical activators. It can accelerate dehydration and depolymerization, promote the hydrolysis of cellulose and hemicellulose and change the structure of lignin. In addition, H_3_PO_4_ cross-linking reactions between H_3_PO_4_ and precursors are conducive to the production of hierarchically porous carbon and hydrophilicity improvement of materials. The preparation of plant-derived carbon materials by H_3_PO_4_ activation, usually mixing carbon precursors or pre-carbonized materials with H_3_PO_4_ and then performing pyrolysis at high temperatures [103,104,105,106]. The application of H_3_PO_4_ activation to regulate plant-based carbon materials is shown in Table 3.

Daoud et al. [112] mixed *Ziziphus jujuba* stones with H_3_PO_4_ as an activator at a ratio of 1:3 and carbonized them at 500 °C to develop carbon materials, which had an outstanding specific surface area (1896 m^2^/g) and extremely high mesoporosity (95.2%). Chatir et al. [109] investigated the impact of the impregnation ratio of H_3_PO_4_ to argan nut shell-derived hydrochar on the activation effect under the pyrolysis temperature of 800 °C. The obtained carbon material showed a specific surface area, a pore volume and mesoporosity up to 1880 m^2^/g, 1.36 cm^3^/g and 50.7%, respectively, when the impregnation ratio was 3. Hu et al. [111] further explored the influences of activation temperature and procedure on the activation effect. In the case of one-step activation, the specific surface area and mesoporosity of lacquer wood-derived activated carbon were superior at 500 °C (Figure 11a–b). However, the material possessed a higher specific surface area and mesoporosity at 800 °C after two-step activation (Figure 11c–d). To sum up, the activation effect of H_3_PO_4_ is closely related to activation procedure and temperature, and using one-step activation at a suitable temperature is more likely to afford carbon materials with the higher specific surface area and mesoporosity. Furthermore, the H_3_PO_4_ activation method is mild with a simple and efficient process, which has a good prospect in the regulation of mesoporosity.

#### 3.2.2. Enzymolysis

Enzymolysis and fungal treatment have similar mechanisms, both of which realize pore size regulation by degrading the composition of plant materials. In comparison to fungal treatment, enzymolysis spends less time on pore size regulation. Researchers have employed enzymolysis to improve the plant-derived carbon material, providing more possibilities for its application in multiple fields. Enzymolysis time and enzyme dosage are critical factors influencing the regulation effect.

Wang et al. [114] used a treatment of wood using cellulase that resulted in a wood-based derived carbon material that was mesoporous dominated; the specific surface area is largest when the treatment temperature and treatment time were 100 °C and 48 h, respectively. Li et al. [115] investigated the effects of enzymolysis temperature and enzyme dosage on the pore size regulation effect (Figure 12). When 100 mg enzyme participated in the reaction at 50 °C, the sample was dominated by mesopores (10–20 nm) with a specific surface area of 1418 m^2^/g, which is 2.6 times that of pristine wood (542 m^2^/g). In short, enzymolysis can effectively improve the pore structure of plant materials, which is capable of producing materials dominated by mesopores under certain conditions. This method provides more possibilities for the pore size regulation of plant-derived carbon materials and opens up new paths for green synthesis.

#### 3.2.3. Molten Salt Activation

In the molten salt activation method, the commonly used molten salts include KCl, LiCl, NaCl, FeCl_3_ and ZnCl_2_. They can be divided by properties into inert salts (KCl, LiCl and NaCl) and active salts (FeCl_3_ and ZnCl_2_). Inert salts own good heat storage and transfer capabilities, which can provide a uniform inert reaction medium for activation. The active salts are able to hinder the agglomeration of particles, thereby further improving the porosity of carbon materials. Particularly, ZnCl_2_ is endowed with the abilities to promote the aromatization of materials during activation and improve the carbon yield [116,117,118]. The preparation of plant-based carbon materials by molten salt activation is shown in Table 4.

Xue et al. [16] mixed rice husks with NaCl/KCl and carried out carbonization of the dried samples at 800 °C. The developed carbon materials presented a large specific surface area of 977 m^2^/g, a pore volume of 0.45 cm^3^/g and mesoporosity of 65.2%. Zuo et al. [125] utilized *Butnea monosperma* pollens for carbon material development and explored the impact of ZnCl_2_ on the pore size. The results indicated that ZnCl_2_ activation successfully prepared carbon materials dominated by mesopores. Lu et al. [120] investigated the impacts of the mass ratio of raw material to ZnCl_2_ on the pore size regulation effect (Figure 13a–c). The results showed that when the mass ratio of pinecone to ZnCl_2_ was 1:4 under the pyrolysis temperature of 600 °C, the specific surface area, pore volume and mesoporosity were up to 1703 m^2^/g, 1.86 cm^3^/g and 90.1%, which enhanced 2.58, 6.91 and 3.66 times compared with that not treated with activation, respectively. Taer et al. [121] studied the pyrolysis temperature impact on the activation effect of banana leaves. According to Figure 13d–f, it could found that banana leaves-derived carbon had a more superior pore structure at the pyrolysis temperature of 800 °C (66.6%) and 900 °C (67.4%). It exhibited a higher specific surface area at 800 °C (860 m^2^/g) compared with 900 °C (626 m^2^/g), which was probably due to the collapse of the pore structure at 900 °C. For the molten salt activation method, activators such as NaCl can be removed directly by hot water immersion. In addition, residual ZnCl_2_ needs dilute acid to remove. Compared with other molten salts, ZnCl_2_ is more commonly used in pore size adjustment. Furthermore, the activation effects of blending two different molten salts still need to be further explored.

#### 3.2.4. Template Method

The template method can be divided into a soft template method and hard template method according to template characteristics. The soft template method mainly uses micelles for decomposition at high temperatures to create pores in materials. For the hard template method, commonly used templates are MgO, SiO_2_, ZnO, MnO, etc. The carbon precursor/pre-carbonized material is usually mixed with a prepared templating agent or a chemical reagent (such inorganic salt solution, KMnO_4_ solution, etc.), which can in situ generate the template agent at high temperatures, and the acid reagent is used to remove the template to obtain the porous carbon material [126]. Table 5 shows the application of the hard template method in the regulation of mesopores.

Du et al. [130] carbonized Na_2_SiO_3_ with carrot at a ratio of 0.2:1 to produce layered carbon material with a high specific surface area (1265 m^2^/g) and mesoporosity (90.5%) at the pyrolysis temperature of 700 °C. Shi et al. [129] investigated the effect of CaCO_3_ on the pore structure of straw-based carbon materials at different pre-carbonization temperatures, activation temperatures and dosages. The results (Figure 14) showed that the specific surface area, pore volume and mesoporosity were maximum when the rice straw was mixed with CaCO_3_ at a ratio of 1:2, which is carbonized at 400 °C and activated at 800 °C. Qiu et al. [132] employed cork as the carbon precursor and KMnO_4_ as the raw material of the MnO template to prepare hierarchically porous carbon and investigated the impact of MnO on the pore structure of the carbon material at different pyrolysis temperatures. It was found that at the pyrolysis temperature of 800 °C, prepared carbon material was dominated by mesopores (60%) and specific surface area up to 1119 m^2^/g. Compared to other methods, the template method is able to regulate the pore size distribution directionally by changing the activator type. Some plants exist with natural templates, such as SiO_2_ in raw husks, which can direct high-temperature pyrolysis and utilize acid to remove SiO_2_, obtaining porous carbon materials dominated by mesopores [41]. CaCO_3_ in apricot shells not only acts as a template in the form of CaO but also can produce CO_2_ at high temperatures to create pores [128]. Such templates are conducive to achieving the high-value utilization of plant materials. However, templates need to be removed by acid, causing the experimental process to be relatively complicated, which restricts the application of the method in practical production.

## 4. Conclusions and Prospects

Plant-based carbon materials are widely used in the fields of electrochemical energy storage, adsorption and catalysis due to their rich chemical composition, more complete carbon skeleton and potential to be transformed into other carbon materials. Among the micropore regulation methods, foam activation, KOH activation and CO_2_/H_2_O(g) activation are more frequently used, while freezing treatment and fungal treatment are less reported. The physical activation method requires a higher temperature and longer time, and the operation is relatively complicated. The KOH activation method and foam activation method are simple and valid methods for enhancing microporosity. Compared to the KOH activation method, foam activation is more environmentally friendly, possessing brilliant application prospects. Regarding mesopore regulation, the H_3_PO_4_ activation and ZnCl_2_ activation are provided to promote aromatization and increase the carbon yield while improving mesoporosity. In comparison, the H_3_PO_4_ activation method requires a lower temperature, which can shorten activation time and effectively improve activation efficiency. The template method is able to regulate the pore size distribution directionally by changing the templating agent but needs a complicated operation. Moreover, according to the existing experimental data, enzymolysis is capable of effectively improving mesoporosity under certain conditions, and the experimental process is relatively simple and non-polluting. Thus, enzymolysis has a good prospect on the pore size regulation of plant-based carbon materials. Based on the above aperture adjustment method, we propose the following perspectives as the focus of future research on the pore regulation of plant-based carbon materials: (1) According to the characteristics of plant materials, develop more green and efficient pore size regulation methods; (2) When adjusting the pore size, consider the influence of the pore regulation method on the pore morphology; and (3) Explore plant-based activators to realize resource recycling and sustainable development.

## Figures and Tables

**Figure 1 polymers-14-04261-f001:**
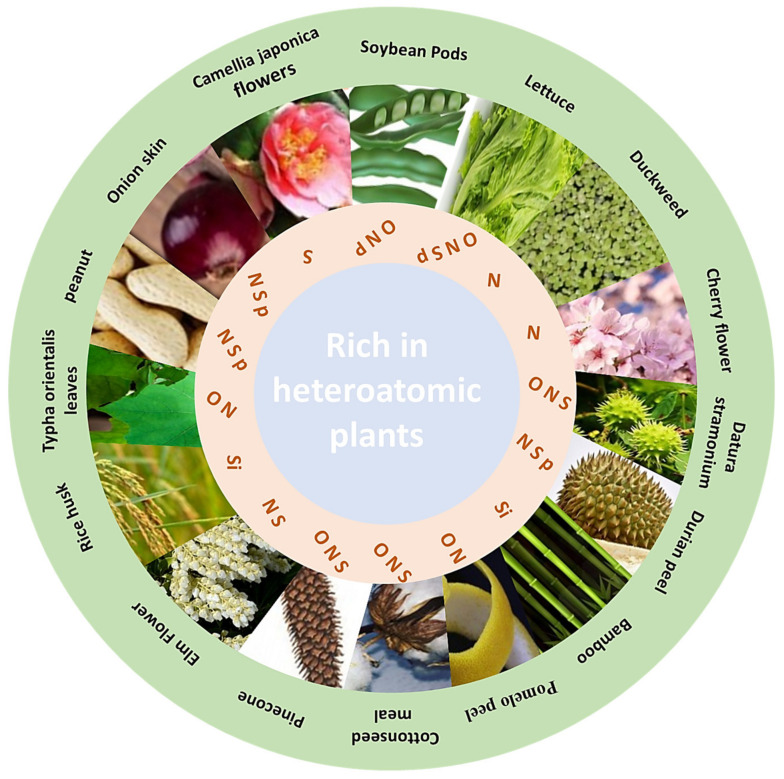
Plants rich in heteroatoms: Elm Flower [37], *Datura stramonium* seed pods [20], bamboo [38], Pinecone [39], Durian peel [40], Rice husk [41], Peanut meal [42], *Typha orientalis* leaves [43], Lettuce slice [44], Soybean pods [45], Onion skin [14], *Camellia japonica* flowers [35], Duckweed [36], Hollyhock leaves [46], Cottonseed meal [47], Pomelo peel [48], Cherry flowers [49].

**Figure 2 polymers-14-04261-f002:**
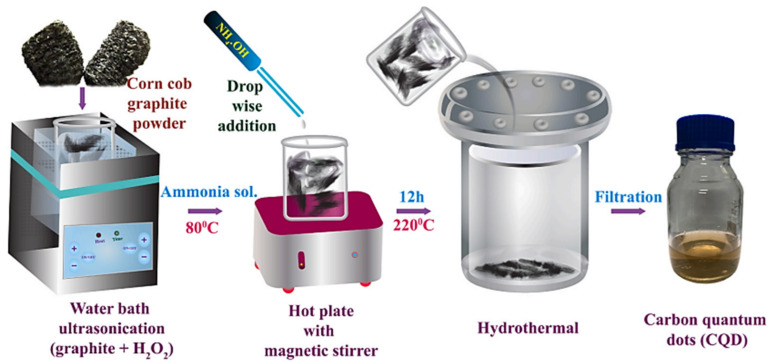
Preparation process of corn cob-derived CQD [52].

**Figure 3 polymers-14-04261-f003:**
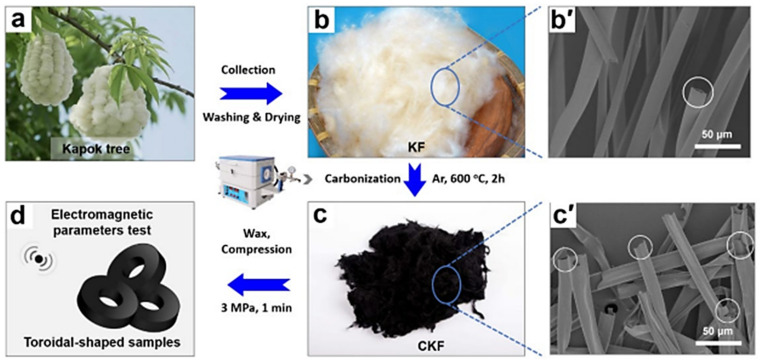
Synthesis process (**a**–**d**) and microscopic morphology of kapok fiber microtubes: uncarbonized (**b’**) and carbonized (**c’**) [56].

**Figure 4 polymers-14-04261-f004:**
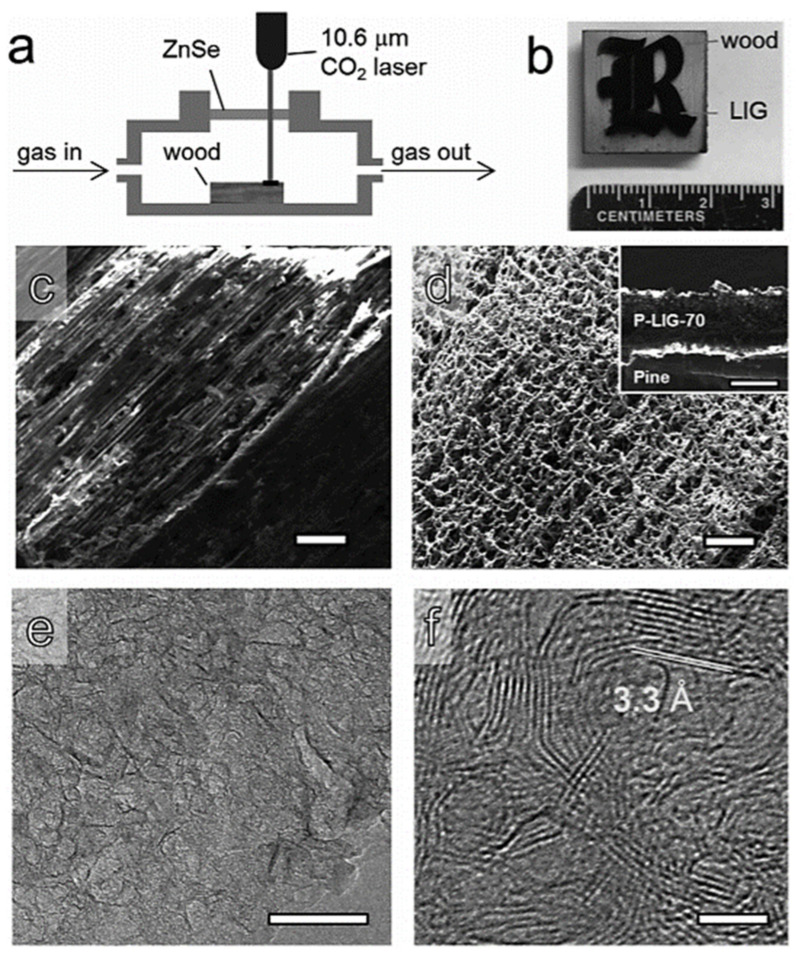
Schematic illustration of laser-induced graphene synthesis (**a**); The pattern of pine etched by the CO_2_ laser (**b**); The SEM (**c,d**) and TEM (**e**,**f**) of P-LIG-70 [62].

**Figure 5 polymers-14-04261-f005:**
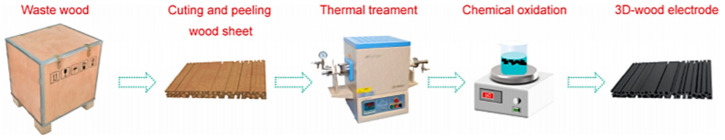
Schematic illustration of 3D-wood electrode synthesis [62].

**Figure 6 polymers-14-04261-f006:**
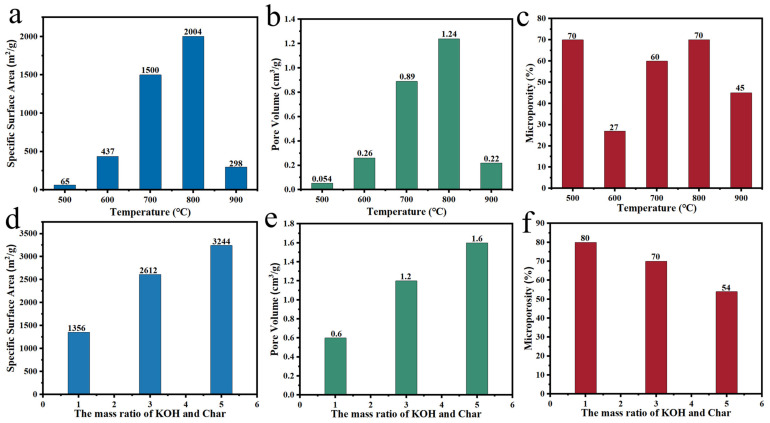
(**a**–**c**) Impact of temperature on KOH activation effect [70]; (**d**–**f**) Impact of char/alkali mass ratio on KOH activation effect [78].

**Figure 7 polymers-14-04261-f007:**
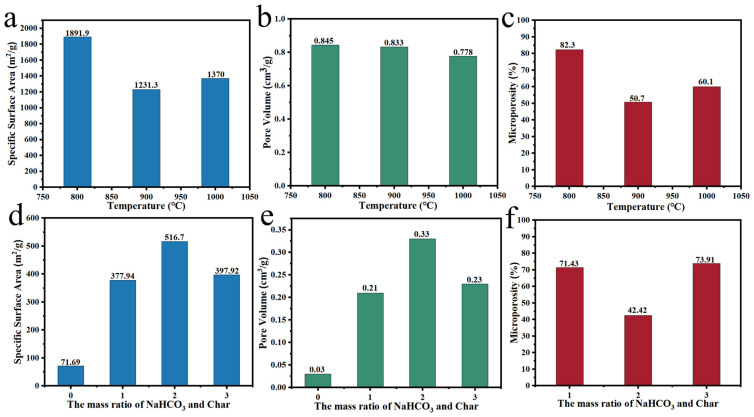
(**a**–**c**) Impact of temperature on NaHCO_3_ activation effect [84]; (**d**–**f**) Impact of char/alkali mass ratio on NaHCO_3_ activation effect [85].

**Figure 8 polymers-14-04261-f008:**
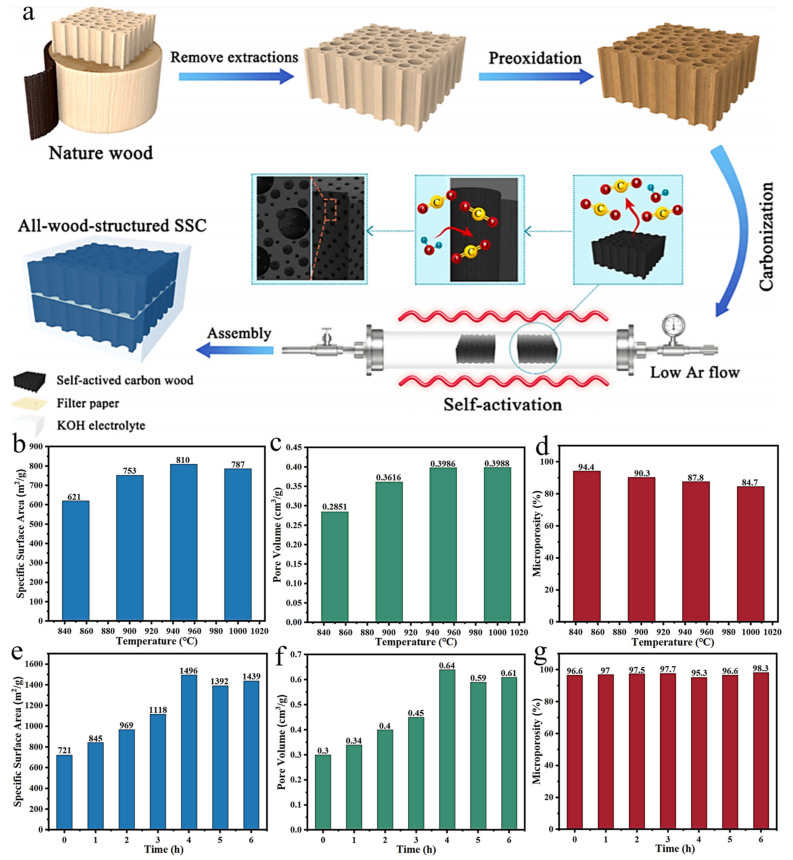
(**a**) Schematic diagram of preparing wood-derived carbon materials by self-activation [95]; (**b**–**d**) Impact of temperature on physical activation effect [89]; (**e**–**g**) Impact of char/alkali mass ratio on NaHCO_3_ activation effect [90].

**Figure 9 polymers-14-04261-f009:**
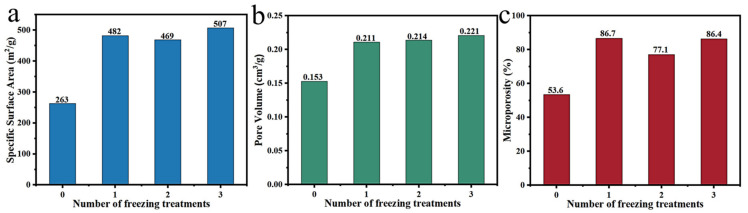
(**a**–**c**) Impact of number of freezing treatments on freeze treatment effect [98].

**Figure 10 polymers-14-04261-f010:**
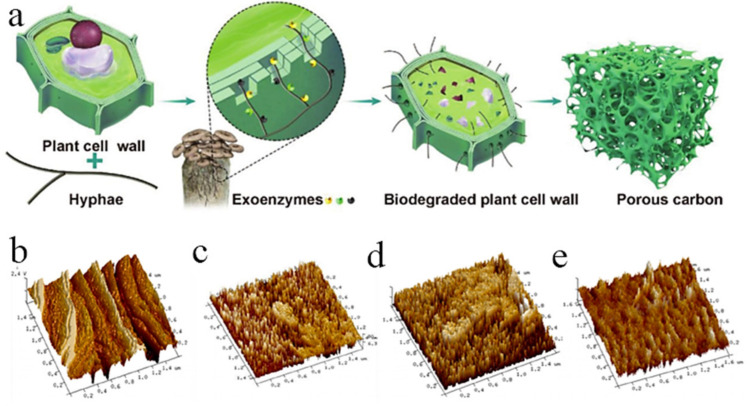
(**a**) Mechanism of pore structure regulation of plant materials by fungi [99]; (**b**–**e**) AFM images of lotus leaves after fungal treatment [100].

**Figure 11 polymers-14-04261-f011:**
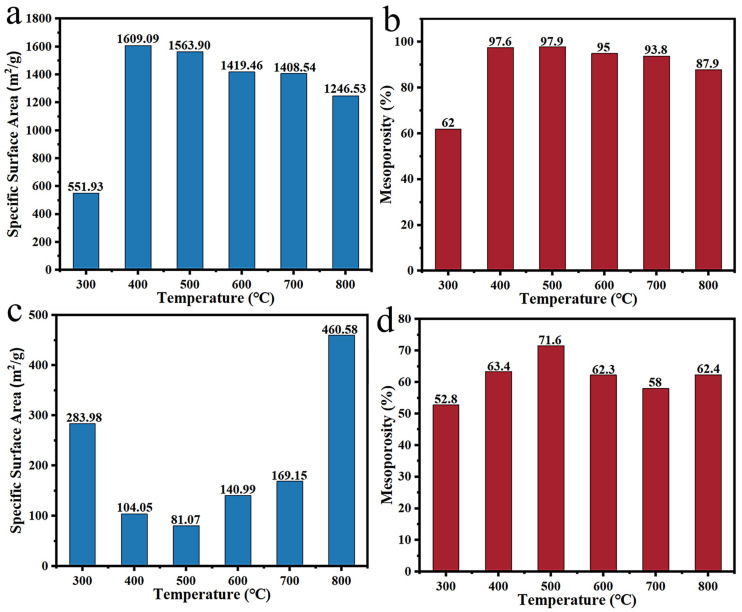
The impacts of one-step activation temperature on specific surface area (**a**) and mesoporosity (**b**); The impacts of two-step activation temperature on specific surface area (**c**) and mesoporosity (**d**) [111].

**Figure 12 polymers-14-04261-f012:**
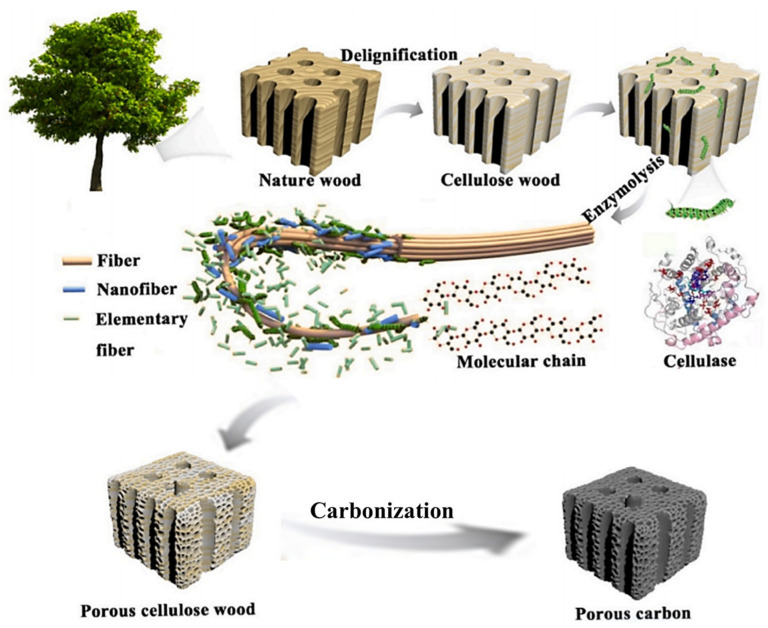
The schematic diagram of enzymatic mechanism [115].

**Figure 13 polymers-14-04261-f013:**
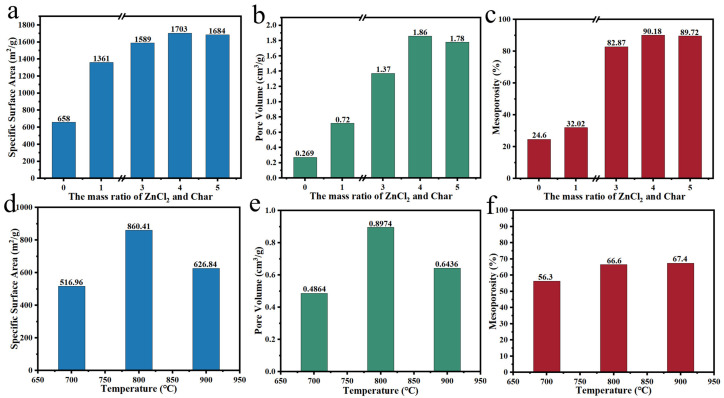
(**a**–**c**) Impact of the mass ratio of ZnCl_2_ and char on ZnCl_2_ activation effect [120]; (**d**–**f**) Impact of the activation temperature on ZnCl_2_ effect [121].

**Figure 14 polymers-14-04261-f014:**
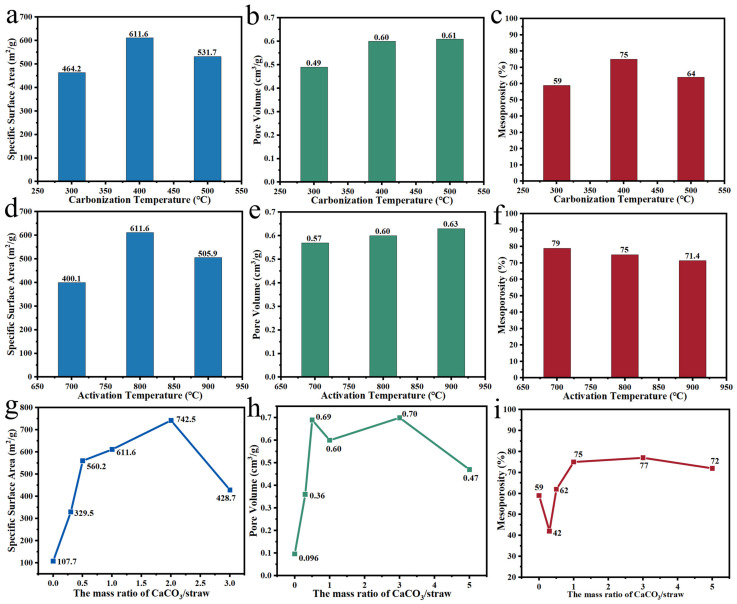
(**a**–**c**) Impact of the carbonization temperature on CaCO_3_ activation effect; (**d**–**f**) Impact of the activation temperature on CaCO_3_ activation effect; (**g**–**i**) Impact of the mass ratio of CaCO_3_/straw on CaCO_3_ activation effect [129].

**Table 1 polymers-14-04261-t001:** Preparation of plant-derived carbon materials by KOH activation.

Material	Method	Temperature (°C)	Ratio	*S*_BET_ (m^2^/g)	*V*_total_ (cm^3^/g)	*V*_mic_*/V*_total_ (%)	Reference
Tea saponin	KOH activation	750	1:1	1550	0.91	62	[69]
Orange peel	KOH activation	800	1:1	2004	1.24	70	[70]
Rice husks	KOH activation	750	1:2	741	0.45	67	[71]
Grapefruit peel	KOH activation	800	1:4.5	3497	1.59	99	[72]
Peanut shells	KOH activation	800	1:4	1272	0.65	52	[73]
Castor shells	KOH activation	800	1:1	891	0.43	85	[74]
Bagasse	KOH activation	800	1:2	1381	0.58	81	[75]
Pine sawdust + sludge	KOH activation	800	1:2	2482	0.89	71	[76]
Rubber seed shells	KOH activation	700	1:3	712	0.36	28	[77]
Biomass waste	KOH activation	850	1:3	2612	1.20	70	[78]

**Table 2 polymers-14-04261-t002:** Preparation of plant-derived carbon materials by physical activation.

Material	Method	Temperature (°C)	Time (h)	Gas Flow Rate (mL/min)	*S*BET (m^2^/g)	*V*total (cm^3^/g)	*V*mic/*V*total (%)	Reference
Olive stone	H_2_O(g) activation	900	0.5	0.15	1191	0.69	75.3	[88]
Bamboo	CO_2_ activation	900	2	-	907	0.44	88.4	[89]
Bamboo	CO_2_ activation	800	4	-	1496	0.64	95.3	[90]
Pine nut shell	H_2_O(g) activation	850	1	18	956	0.62	62.9	[91]
Coconut shell	H_2_O(g) + CO_2_ activation	900	6	-	1194	0.53	87.8	[92]
Barley straw	CO_2_ activation	800	1	2500	789	0.35	93.5	[93]
Corn cob	H_2_O(g) activation	800	1	-	566	0.32	89.0	[94]
Apricot shell	CO_2_ activation	500	5	10	1690	0.93	52.6	[87]
Wood	CO_2_ activation	1000	10	10	1145	0.47	50.3	[95]

**Table 3 polymers-14-04261-t003:** Preparation of plant-derived carbon materials by H_3_PO_4_ activation.

Material	Method	Temperature (°C)	Ratio	*S*_BET_ (m^2^/g)	*V*_total_ (cm^3^/g)	*V*_mes_/*V*_total_ (%)	Reference
Bitter orange peel	H_3_PO_4_ activation	550	1:1	611	0.57	92.4	[107]
Poplar catkins	H_3_PO_4_ activation	600	1:3	2001	1.22	91.7	[105]
Spent mushroom substrate	H_3_PO_4_ activation	900	1:2	1215	0.83	87.4	[108]
Argan nut shell	H_3_PO_4_ activation	500	1:3	1880	1.36	50.7	[109]
Chinar fruit fluff	H_3_PO_4_ activation	600	-	1759	1.68	97.4	[110]
Lacquer wood	H_3_PO_4_ activation	400	-	1609	1.46	97.6	[111]
*Phoenix dactylifera rachis*	H_3_PO_4_ activation	500	1:3	1283	1.72	83.5	[112]
*Ziziphus jujuba* stones	H_3_PO_4_ activation	500	1:3	1896	1.26	95.2	[112]
Kenaf	H_3_PO_4_ activation	500	1:1	1510	1.26	57.0	[113]

**Table 4 polymers-14-04261-t004:** Preparation of plant-derived carbon materials by molten salt activation.

Material	Method	Temperature (°C)	Ratio	*S*_BET_ (m^2^/g)	*V*_total_ (cm^3^/g)	*V*_mes_/*V*_total_ (%)	Reference
Willow leaves	ZnCl_2_	800	1:3	809	0.98	95.5	[119]
Pinecone	ZnCl_2_	600	1:4	1703	1.86	90.1	[120]
Banana leaves	ZnCl_2_	800	-	860	0.89	66.6	[121]
Rice husks	NaCl/KCl	800	-	977	0.45	65.2	[16]
Tree bark	ZnCl_2_	900	1:1	1114	0.78	47.4	[122]
Cornstalk	NaCl/KCl	800	1:1	864	1.01	72.8	[123]
Cornstalk	LiCl/ZnCl_2_	800	1:1	1276	0.80	78.9	[124]
Bitter orange peel	ZnCl_2_	450	1:1	1450	0.78	87.3	[107]
*Butnea monosperma* pollens	ZnCl_2_	800	1:2	1422	0.76	57.3	[125]

**Table 5 polymers-14-04261-t005:** Preparation of plant-derived carbon materials by hard template method.

Material	Method	Temperature (°C)	*S*_BET_ (m^2^/g)	*V*_total_ (cm^3^/g)	*V*_mes_/*V*_total_ (%)	Reference
Plane tree bark	ZnO	900	1587	2.33	82.8	[127]
Apricot shells	CaO	700	931	1.48	73.6	[128]
Rice straw	CaO	800	742	0.70	77.1	[129]
Corn stalk	MgO	600	218	146.00	34.0	[125]
Carrot	SiO_2_	700	1265	2.10	90.4	[130]
Cork	MnO	800	1199	0.98	60.0	[131]
Rice straw	MgO	500	158	0.16	42.0	[132]

## Data Availability

Not applicable.

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
