# Peer review of "Advances in Micro-/Mesopore Regulation Methods for Plant-Derived Carbon Materials"

_polymers, 2022, doi:10.3390/polym14204261_

Round 1
Reviewer 1 Report
This paper introduces brief insight into the methods of biomass activation. While being quite comprehensive it barely adds anything new to the existing knowledge, and also offers some dubious statements, which I will discuss below.
English should be improved, some sentences are incorrect or awkward, few examples:
Novel carbon materials mainly include graphene [2], carbon nanotubes [3], porous carbon [4], and fullerene [5], which can be prepared by resins [6], gels [7], biomass materials [8], and petroleum pitch [9]. – should be “from”.
Biomass-based carbon materials can be obtained by high-temperature treatment (hydrothermal carbonization, pyrolysis, and gasification) under oxygen-limited conditions, which have good electrical conductivity and high porosity – wrong sequence of words. Should be rather “Biomass-based carbon which have good electrical conductivity and high porosity materials can be obtained by high-temperature treatment (hydrothermal carbonization, pyrolysis, and gasification) under oxygen-limited conditions”.
The hydrophilicity of -OH is conducive to promoting the contact of carbon materials and aqueous solution, endowing the materials with more characteristics. – awkward construction
Etc.
(1) Alkali solution is pyrolyzed at high temperatures to produce CO2, CO, etc., which can etch carbon to form more pores; - this is not the actual mechanism. I would suggest book my Henry Marsh where all these mechanisms are extensively studied.
The K atom dissociated from KOH can embed into all materials, while the Na atom sourced from NaOH is only for ordered materials. – not true, we have more success using NaOH for biomass activation than in the case of KOH. In either case both alkalis are suitable for the task.
Table 1 – why two lines are in yellow?
However, the corrosiveness of KOH and its harm to the environment limit its application in commercial production – this is not the main reason. KOH is not corrosive to metal and K salts are important and needed for agriculture.
The porous carbon had a specific surface area of 397.92 m2/g and the microporosity of 73.91% at the mass ratio of 1:3 (CF:NaHCO3). – this is not a great result at all!
While using this method, researchers should take into account the activation temperature and activator dosage for a more ideal activation effect. – this is true for all activation methods.
To sum up, the activation effect of H3PO4 is closely related to activation procedure and temperature, and one-step activation and a suitable temperature are more likely to afford carbon materials with an ideal pore structure. – what is “ideal pore structure” for carbons?
It should be noted that the samples obtained by the molten salt activation method need to be soaked in water or an enolic acid solution to remove the activator – washing is an obligatory step for all chemical activation methods.
Reviewer 2 Report
This review should be accepted in present form.
Reviewer 3 Report
The manuscript written by J. Liu et al. describes an interesting short review on plant-derived carbon porous materials and methods of control micro- and mesoporosity. The number of presented carbon precursors is typical of rather short review, which this manuscript represents. The provided information is logically presented and comprehensive.
I recommend a minor revision before acceptance.
1. I strongly suggest to add more graphs like one in Figure 7, to sum up each paragraph (each method) and comment more relationships between conditions of preparation and textural parameters (e.g. temperature of carbonization vs SBET). The title of the manuscript is about regulation methods, therefore these methods should be presented. Showing only one, specific example from particular paper is not sufficient, the correlation found by authors is more important to be shown.
2. Figures 7 and 10 are too small, especially font size is unreadable. This comment refers also to other Figures. I suggest to check all of them in terms of readability.
3. In paragraph 5.2.4. Template method, the description of hard template method is misleading. The sentence suggest, that ZnO, MnO or SiO2 are compounds which are mixed with carbon precursors. In fact, compounds which are mixed with carbon precursors are inorganic salts, which decompose during heat treatment into ZnO, MnO or SiO2. Examples are presented later in the text, but please rewrite the sentence.
4. The name of Table 5 should be more specific, i.e. … carbon materials by hard template method.
5. Some of references do not contain DOI numbers.
Round 2
Reviewer 1 Report
Accept in present form